# Inter e intra-variability of the best ranked teams: A network analysis in male high-level volleyball

**Augusto Cezar Rodrigues Rocha[1], Lorenzo Laporta[2], Geovana Pires Rodrigues[1], Juracy da Silva Guimarães[1], Marcos Henrique do Nascimento[1], Marcelo Couto Jorge Rodrigues[1], Thiago José Leonardi[3], Claudio Andre Barbosa de Lira[1], Henrique de Oliveira Castro[4], Gustavo De Conti Teixeira Costa[1]** *

1 Núcleo de Estudos e Pesquisa Avançada em Esportes (NEPAE), Universidade Federal de Goiás, Goiânia, Brazil, 2 Núcleo de Estudos e Pesquisa Avançada em Esportes (NEPAE), Centro de Educação Física e Desportos, Universidade Federal de Santa Maria, Santa Maria, Brazil, 3 Laboratório de Estudos Multidisciplinares em Esportes, Escola de Educação Física, Fisioterapia e Dança, Universidade Federal do Rio Grande do Sul, Porto Alegre, Brazil, 4 Núcleo de Estudos e Pesquisa Avançada em Esportes (NEPAE), Universidade Federal do Mato Grosso, Mato Grosso, Cuiabá, Brazil

* conti02@ufg.br

**Data Availability Statement:** All relevant data are within the manuscript and its Supporting Information files.

## Abstract

The present research objective was to analyze the offensive phase from Complex I in high-level male volleyball teams in a macro- and micro-level view, through the inter e intra-team variability analysis of eight best teams of the 2018 Men's Volleyball World Championship over the social network analysis and eigenvector centrality. The sample consisted of 22 matches and 2,743 offensive actions, resulting in 8 sub-networks with 368 nodes and 6221 edges. The results showed from macro view the variables that presented highest centrality values were Attack Zone 4 (range 0.56–0.90), Attack Tempo 2 (0.65–0.87), Power Attack (0.62–0.94), No Touch Block (0.61–1), Attack Effect Continuity (0.59–0.94), and Middle Blocker Centralized (0.60–0.95). In a micro view, Reception Effect, Play Position, Reception Zone, and Block Composition showed high variability in each sub-network. The intra- and inter-team variability presented the importance of to respect each team idiosyncrasies and to consider the different approaches to the game and success.

## Introduction

Performance Analysis uses research, training and competition understanding for practical applicability in Teams Sports. In turn, Match Analysis (MA) allows the evaluation and characterization of tactical-technical actions in different contexts [1]. In this context, team sports analysis must consider the sports specificity and complexity in their action context, that is, respecting the game ecology [2], considering the interactions between the actions and game environment [3, 4]. In this perspective, the possibilities of action observed by individual interactions with environmental restrictions are known as affordances [5], the context being essential for having interactions between the individual and the environment, providing

**Funding:** The author(s) received no specific funding for this work.

**Competing interests:** The authors have declared that no competing interests exist.

opportunities for behaviors adaptations to solve emerging problems during competitive performance [6]. Therefore, the environment is described from the perceptions and possible actions in the different environmental conditions that constitute the game actions [7].

In volleyball, the game context can be analyzed from interconnected game actions and influenced by subsequent actions [8–10]. Game Complexes (K) are composed of game actions, and are essentially characterized by Complex I (KI) consisting of serve-reception, setting and attack, while Complex II (KII) is constituted by the serve, blocking, defense, setting and counterattack [11–13]. Analyzing the game from the Game Complexes perspective becomes important when considering the actions behavior in each phase, being in-system or in ideal conditions for offensive construction, or off-system with limited conditions for offensive construction [14, 15].

Research on volleyball in Complex I considered that reception predicts the setting efficacy [9, 16] and the attack success (point) [17–19], and when the reception is not effective (decreased attack options) the chances increase of lost the match [20, 21]. Reception also influences the setting location revealing the importance of subsequent actions [22]. There are high possibilities of in-system situations occurring in this complex [23, 24], with more powerful attacks occur after fast settings resulting in attack points [25]. On the other hand, actions off-system increase the blockers number, with slower settings and reduce the scoring points chances in the attack [14, 26]. Thus, teams obtain more attack points in KI [27, 28], because of the greater predictability of the serve-reception, and, consequently, better ball control and offensive structuring [29–31].

Although the game patterns described above are well established in the literature, the most derived from inferential and predictive analyzes [32, 33], not considering the relationship between the variables analyzed in a global way. Thus, when considering the game as a dynamic system, constantly modified based on the occurance actions [10, 34], it becomes necessary to interpret the game within its context, considering its ecology [35], specificities, and avoiding generalizations [36].

Considering the Teams Sports complexity, direct and indirect relations to game actions should be considered, according to the performance context [14]. In this context, Social Network Analysis (SNA) is a tool that allows understanding the game complexity, expanding the individual analysis of the fundamentals, which may be insufficient to understand what happens within this complex and dynamic system that is the volleyball game [37]. Studies about men's high-level volleyball based on SNA reveal that most teams play in-system in KI, and at the same time, present in- and off-system situations in the offensive construction in KII [10, 14, 15, 38]. In addition, when considering only in-system situations in KI, the reception zones further away from the network (Z5, Z6, Z7, Z8, Z9 and Z1) presented higher centrality values with the other game actions, suggesting greater diversity in the game type played [22].

Furthermore, research on the match analysis of high-level volleyball answered questions about the game patterns in determined competitions, understanding little about intra- and inter-team variability pattern of each team. Considering that affordances are inherent to the game context and that the offensive construction of each team occurs according to the specific team demands, from that, the study problem consisted in to contribute from the game patterns investigate respecting each team individuality (intra-team variability), game ecology and the action possibilities created by the teams according to specific game scenarios, respecting the coexistence of multiple performance models (inter-team variability) [39]. In addition, despite the ecological character respected by the SNA, methodologically the volleyball studies analyzed each selection in a separate network, which can provide different results when analyzing the individual standards of each performance model.

Thus, the objective was to consider the multiple models of high performance in volleyball, investigating at two levels of analysis: (a) macro-level analyzing the important variables for the

construction of the KI of the eight best ranked teams that participated in the 2018 Volleyball World Championship through a single and global competition network, and (b) micro-analysis considering the peculiarities (differences and similarities) in the offensive construction in KI that distinguish the multiple performance models of the teams participating in the competition. Therefore, from the SNA, we hypothesize that: (a) in macro-analysis view the teams will present higher centrality values to excellent setting conditions (reception effect A), attack tempo 1, Attack from Zone 3, against simple block (1x1), with attack does not touch block and attack effect point, and (b) in micro-analysis the top 2 ranked teams will present higher eigenvector values in the reception effect, attack and blocking effectivity.

## Materials and methods

### Sample

A total of 2,743 offensive actions in Complex I were analyzed, from 22 matches of 2018 Men's Volleyball World Championship between the eight best ranked teams in this edition (Poland, Brazil, United States, Serbia, Italy, Russia, France and Netherlands respectively). A global network was created containing the information of each team, consisting in 8 sub-networks with 368 nodes and 6221 edges. The Ethics Committee at Universidade Federal de Goiás provided institutional approval for this study under the CAAE protocol 15137319.6.0000.5083

### Variables

**Reception Zone (RZ).** Was defined from 9 zones (RZ's) with dimensions of 3m x 3m where the reception actions took place [22, 31]. When the athlete had one foot in one zone and the other foot in another zone, the zone where the ball (or most of it) was located at the time of reception was considered. RZ6, for example, indicates reception in zone 6 (See Fig 1). *Reception Effect (RE)* was determined from the relationship between reception and available attack options, as follows: REA—all available attack options; REB–fast attacks are possible, but with more difficulty and some attack combinations are inhibited; and REC—slow attacks and attacks to the ends of the network are more likely to occur [15, 40].

**Attack Tempo (AT).** Was considered as: AT1- the attacker is in the air or jumping at the moment of setting; AT2—the attacker takes one or two steps after setting; and 3 –the attacker takes three or more steps after setting. *Attack Type (ATT)* was adapted from Costa, Ferreira [41], being: Powerful Attack (ATT POW) an attack performed with power on the ball giving a descending trajectory; Placed Attack (ATT PLA) the ball is attacked with controlled application of force and directed into a vulnerable defensive area; Fingertips Attack (ATT TIP) the ball is contacted with fingertips and directed into a vulnerable defensive area; and attack others (ATT OTH).

**Play position.** We consider the functional player specialization. According to Sheppard, Gabbett [42] the play positions are setter (SET), outside-hitter, middle-blocker, opposite (OPP), and libero (LIB). However, for the present study, we considered outside hitter 2 (H2) and middle-blocker 2 (M2) as the closest attackers to the setter, while outside-hitter 3 (H3) and middle-blocler 3 (M3), were the most distant attackers from the setter.

**Attack zone.** The distribution of the attack positions was used according to the official rules published by the International Volleyball Federation. Since there was no attack at position 5 (mostly used by the libero), we used the following descriptions: Position 1 (AZ1): located between the right sideline, the end line, the attack line (3-meter line) and three meters to the left of the right sideline; Position 2 (AZ2): located between the right sideline, the centerline, the attack line (3-meter line) and three meters to the left of the right sideline; Position 3 (AZ3): located between 3m of the right sideline, centerline, the attack line (3-meter line) and 3 meters

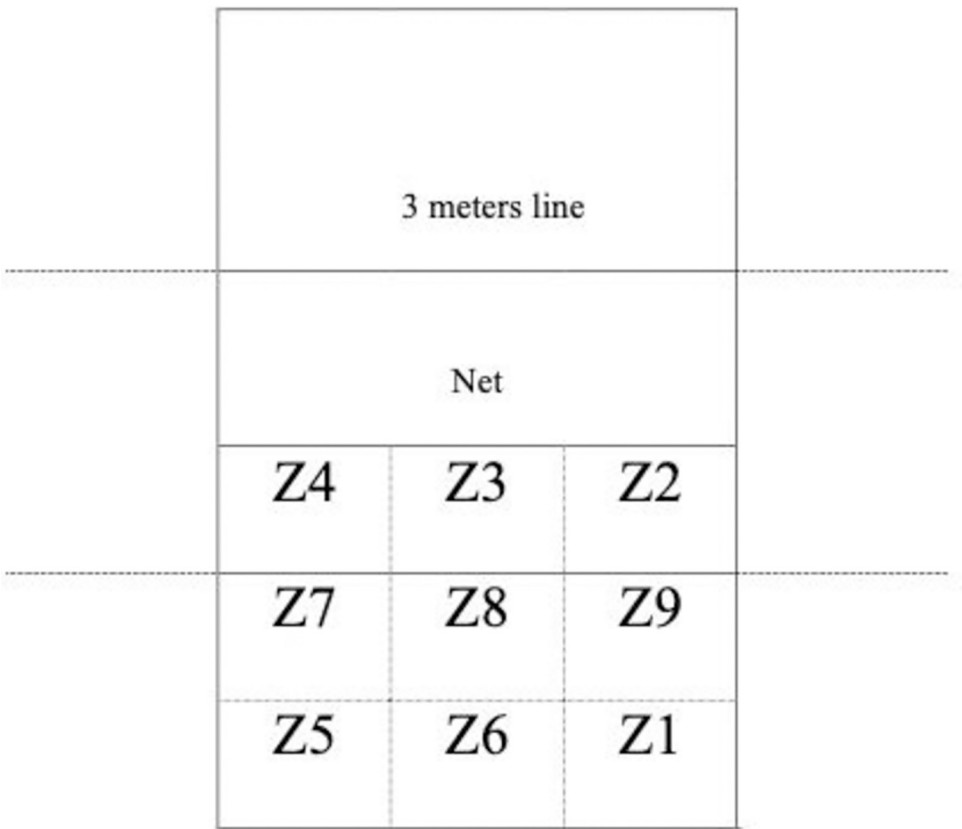

**Fig 1. Reception zones.**

to the right of the left sideline; Position 4 (AZ4): located between the left sideline, the center-line, the attack line (3-meter line) and three meters to the right of the left sideline; Position 6 (AZ6): located between 3m of the right sideline, the end line, the attack line (3-meter line) and 3 meters to the right of the left sideline.

**Block composition.** Was considered the game ecology and the influence of subsequent actions to KI. Therefore, opponent blocking was analyzed as it interferes with the attacker behavior and the offensive construction in KI. Blocking were classified as: Triple (1x3), Triple Broken (1x(2 + 1)); Double (1x2), Double Broken (1x (1 + 1)), Single [1x1], No Block by Setter Merit (NO BLOCK +); and No Block by Setter Error (NO BLOCK -) [43]. *Block Touch* as computed with the attack contact by the block, as follows: the ball attacked Touched the Block (Block-T) or the ball attacked did not touch the block (Block-NT). Block Tactical Organization: Block organization refers to the team's tactical commitment to the game's defensive strategies. Thus, we consider that the central blocker could: 1 –stay centered on the net to read the game and then move to the block (B-Center), 2-–anticipate the opponent middle-blocker (B-MH), 3 –anticipate the positioning for zone 4 (B-Z4), and 4 –anticipate the positioning for zone 2 (B-Z2). Such an interpretation of block positioning is not uncommon and demonstrates the type of strategy used by the team to reduce the chances of success of the opponent's attack (For more information, see Afonso and Mesquita [44], Costa and Maia [45]).

**Attack effect.** Was classified as: Error—the ball was attacked into the net, out or violates the regulation; Blocked—the attacker fails due to the opponent blocking; Continuity the attack was defended and allows the counterattack; Point—point—the attack results in a direct point [46].

## Data collection

The games were filmed in high-definition format (1080p) in behind view of the court (7 to 9m) and above ground level (5m). The analysis was performed by three volleyball coaches (more than five years of experience as performance analysts), trained by a high-level coach with more than 10 years of practice and with experience with national and regional teams. The coaches used the Data Volley software to control and analyze the scenes. Each analyzed dimension was discussed by the coaches, ensuring that the categories were exclusive and exhaustive to represent the different game scenarios, as well as that there was a unanimous consensus among all evaluators as suggested by Pulido et al. [47]. For this purpose, 5 games were analyzed together from a competition different from the sample (Final phase of the Men's Brazilian Superliga 20/21), and the final game (composed of 5 sets) was reanalyzed after 1 month to verify the intra and inter-observer reliability, thus resolving any doubts. For the final reliability test, 20% of the actions were reanalyzed, which is above the reference value of 10% [48]. Cohen's Kappa values were between 0.89 and 0.98 with the respective standard errors of 0.07 and 0.01 for the intraobserver analysis, and 1 with the standard error equal to 0 for the interobserver analysis. Such values are above the value recommended by the literature, which is 0.75 [49]. Reliability values for each variable were: Reception Zone (0,95), Reception Effect (0,89); Attack Tempo (0,95), Attack Type (0,98), Attack Zone (1,00), Play Position (1,00), Block Composition (0,98), Block Touch (0,98), and Attack Effect (0,98).

## Data analysis

The analyzed data were recorded in a spreadsheet (Microsoft Excel 2015) enabling data quality control, and later analyzed in IBM SPSS Statistics (Version 27, USA), to perform exploratory statistics in cross tables. For the Social Network Analysis, Gephi 0.9.2-beta software (Version 10.16, France) was used to examine the connectivity and specificity of the relationships between the game variables, through the eigenvector centrality analysis of the variables for each team. Thus, the eigenvector centrality provides the relevant information about which nodes, or here as game variables used, are more influential in the network, taking into account the connectivity from other nodes that are also more central [50, 51], so, the Eigenvector Centrality depends not only on the number of its adjacent nodes, but also their interaction characteristics [40]. Node size were manipulated to highlight the magnitude of the eigenvector measure using the intrinsic units provided by Gephi Software (Between 300 to 1,500 arbitrary units). Edge corresponds the direct relationship between two nodes defined by number of connections, therefore, thicker edges correspond to a greater number of connections between two nodes [10]. Thus, the node size determines the visual variables contrast according to the eigenvector centrality and the edges thickness, in turn, reveals the weight (given by the number) of the connection directly and indirectly between the nodes [10, 24]. In addition, the Modularity Algorithm was used to detect community structure (8 communities with a cohesion value of 0.871) and the "Fruchterman Reigold" distribution (area 100000) was used organizing the nodes with the highest eigenvector centrality in the center of each subnet [52].

## Results

A global network showing intra- and inter-team variability was established through interactions between the eight teams resulting in 8 sub-networks (See Fig 2 and Table 1). The graph distribution organizes the variables with greater centrality in the center of each sub-network.

From a global macro view (See Fig 2), the variables that presented the highest centrality values in the KI offensive organization were: Attack Zone 4 (range 0.45–0.90), Attack Tempo 2 (range 0.45–0.91), Powerful Attack (0.47–0.94), Block Not Touch (0.49–1), Attack Effect

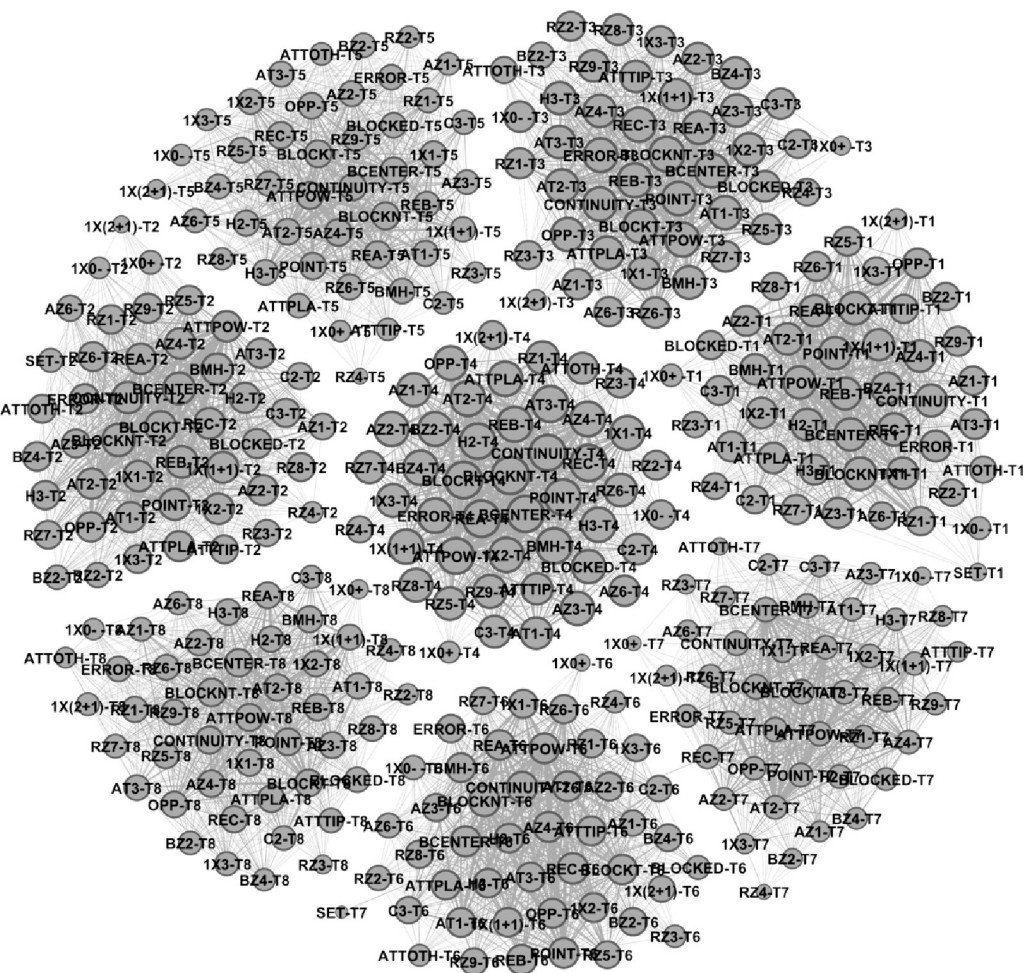

**Fig 2. Global network with inter-team variability.** Each team was represented with your classification number (1 means the first place or Team 1).

Continuity (0.48–0.94) and Blocker in the Center (0.48–0.95). On the other hand, the variables Reception Effect, Play Position, Reception Zone, Block Composition presented a variability in each subnetwork and will be analyzed below.

From a micro-analysis, team 1 (Poland) presented eigenvector values (See Fig 3A and Table 1) for Reception Zone 9 (0.69) and 7 (0.69), Reception Effect C (0.81), Attack Zone 4 (0.78) and Attack Tempo 2 (0.77) and 3 (0.71), with Outside Hitter 2 (0.79) and Powerful Attack (0.81) and Attack Effect Point (0.80) showing higher centrality values in each category. Block Structure Centralized (0.81), with 1x1 (0.77) and 1x(1+1) (0.76), Block not Touch (0.82) were more important in each category in the KI. Team 2 (Brazil–See Table 1 and Fig 3B) revealed higher eigenvector values for Reception Zone 9 (0.69) and 6 (0.67), Reception Effect B (0.78) Attack Zone 4 (0.73), Attack Tempo 2 (0.73) and 1 (0.72), Outside Hitter 2 (0.72) and Opposite (0.69), Powerful Attack (0.73) and Attack Effect Continuity (0.77). Block Structure Centralized (0.78), 1x(1+1) (0.75) and Block not Touch (0.81) showed high centrality values.

Team 3 (USA) is shown in the Graph below (See Fig 4A) presented higher centrality values for each category (Table 1) for Reception Zone 9 (0.77), with the same value of Reception Effect B and C (0.88 for both), Attack Zone 4 (0.81) and 3 (0.81) with Attack Tempo 2 (0.84) and 1 (0.83), Opposite Player (0.85) and Powerful Attack (0.87) and Attack Effect Error (0.86)

**Table 1. Eigenvector values for each team.**

| Variables | 1 | 2 | 3 | 4 | 5 | 6 | 7 | 8 |
|---|---|---|---|---|---|---|---|---|
| RZ1 | 0.697374 | 0.676376 | 0.777922 | 0.834238 | 0.53525 | 0.706419 | 0.545906 | 0.631136 |
| RZ2 | 0.621148 | 0.510327 | 0.702454 | 0.671802 | 0.449045 | 0.548878 | 0.00 | 0.407298 |
| RZ3 | 0.581047 | 0.589313 | 0.678029 | 0.706565 | 0.368088 | 0.445121 | 0.357855 | 0.309162 |
| RZ4 | 0.599749 | 0.364698 | 0.613969 | 0.57157 | 0.27716 | 0.444161 | 0.148795 | 0.473159 |
| RZ5 | 0.710818 | 0.680833 | 0.750143 | 0.820483 | 0.609986 | 0.654492 | 0.529399 | 0.64765 |
| RZ6 | 0.725884 | 0.689377 | 0.761523 | 0.826801 | 0.624964 | 0.712201 | 0.545906 | 0.616465 |
| RZ7 | 0.734263 | 0.645538 | 0.771898 | 0.783562 | 0.57038 | 0.6439 | 0.460394 | 0.546253 |
| RZ8 | 0.657081 | 0.650072 | 0.762353 | 0.837571 | 0.484098 | 0.685969 | 0.402572 | 0.609186 |
| RZ9 | 0.733195 | 0.704697 | 0.80812 | 0.837571 | 0.594266 | 0.650377 | 0.49078 | 0.618101 |
| REA | 0.821706 | 0.755558 | 0.871817 | 0.975953 | 0.690076 | 0.769556 | 0.553726 | 0.652684 |
| REB | 0.846875 | 0.80917 | 0.918335 | 0.910976 | 0.606006 | 0.776886 | 0.595939 | 0.683466 |
| REC | 0.868019 | 0.775171 | 0.918335 | 0.965396 | 0.673292 | 0.807964 | 0.582104 | 0.706387 |
| AT1 | 0.761726 | 0.7553 | 0.86558 | 0.781839 | 0.629143 | 0.739888 | 0.537583 | 0.63781 |
| AT2 | 0.826188 | 0.756769 | 0.877446 | 0.91073 | 0.658097 | 0.779152 | 0.56016 | 0.680843 |
| AT3 | 0.7627 | 0.740878 | 0.846432 | 0.955301 | 0.618832 | 0.731263 | 0.566039 | 0.619163 |
| ATTPOW | 0.862113 | 0.772967 | 0.912749 | 0.943882 | 0.682111 | 0.775901 | 0.58363 | 0.705813 |
| ATTPLA | 0.7898 | 0.746058 | 0.878057 | 0.891187 | 0.565465 | 0.739005 | 0.599713 | 0.689931 |
| ATTTIP | 0.69176 | 0.65528 | 0.810536 | 0.760038 | 0.448851 | 0.746295 | 0.409533 | 0.565658 |
| ATTOTH | 0.607763 | 0.665848 | 0.592132 | 0.756555 | 0.507402 | 0.422123 | 0.254078 | 0.383897 |
| H2 | 0.840587 | 0.747979 | 0.828575 | 0.927845 | 0.614013 | 0.733394 | 0.560612 | 0.635028 |
| H3 | 0.756012 | 0.683398 | 0.835196 | 0.891206 | 0.611991 | 0.72671 | 0.505045 | 0.643504 |
| C2 | 0.660403 | 0.594792 | 0.6635 | 0.750108 | 0.453446 | 0.614565 | 0.436081 | 0.564189 |
| C3 | 0.724636 | 0.673393 | 0.759826 | 0.791887 | 0.528633 | 0.642423 | 0.457762 | 0.496774 |
| OPP | 0.738071 | 0.742799 | 0.883474 | 0.863464 | 0.6417 | 0.741583 | 0.543043 | 0.647331 |
| SET | 0.227414 | 0.00 | 0.00 | 0.00 | 0.00 | 0.00 | 0.00 | 0.00 |
| AZ1 | 0.703297 | 0.584236 | 0.777123 | 0.843669 | 0.48214 | 0.593731 | 0.428612 | 0.54078 |
| AZ2 | 0.76166 | 0.74867 | 0.795376 | 0.885539 | 0.616827 | 0.708106 | 0.518367 | 0.641991 |
| AZ3 | 0.77514 | 0.66224 | 0.840656 | 0.842088 | 0.550643 | 0.676516 | 0.481044 | 0.661973 |
| AZ4 | 0.828728 | 0.762075 | 0.845913 | 0.906619 | 0.686327 | 0.78633 | 0.562201 | 0.685168 |
| AZ6 | 0.631175 | 0.567761 | 0.715851 | 0.732492 | 0.551443 | 0.549251 | 0.464582 | 0.525983 |
| 1X1 | 0.824008 | 0.754607 | 0.854346 | 0.873012 | 0.64227 | 0.715937 | 0.553291 | 0.656835 |
| 1x(1+1) | 0.807178 | 0.768799 | 0.831004 | 0.891769 | 0.596194 | 0.728668 | 0.544369 | 0.620722 |
| 1X2 | 0.789165 | 0.752623 | 0.854346 | 0.891769 | 0.550559 | 0.745931 | 0.567334 | 0.638041 |
| 1x(2+1) | 0.354884 | 0.192167 | 0.352787 | 0.636962 | 0.332101 | 0.61921 | 0.426836 | 0.489819 |
| 1x0+ | 0.436883 | 0.454616 | 0.277898 | 0.381224 | 0.402678 | 0.196904 | 0.149816 | 0.319361 |
| 1x0- | 0.441903 | 0.378565 | 0.770275 | 0.660171 | 0.496842 | 0.526249 | 0.282303 | 0.436208 |
| 1X3 | 0.647315 | 0.645638 | 0.765999 | 0.834889 | 0.457038 | 0.645556 | 0.401286 | 0.538177 |
| BLOCKT | 0.871146 | 0.781418 | 0.908431 | 0.971362 | 0.695398 | 0.767521 | 0.593914 | 0.698496 |
| BLOCKNT | 0.873238 | 0.829896 | 0.949646 | 1 | 0.702804 | 0.821334 | 0.616344 | 0.739621 |
| BCENTER | 0.863573 | 0.802928 | 0.902972 | 0.954233 | 0.663561 | 0.793329 | 0.60199 | 0.729568 |
| BMH | 0.794839 | 0.778921 | 0.857476 | 0.926245 | 0.625913 | 0.740739 | 0.585078 | 0.66253 |
| BZ4 | 0.813503 | 0.677416 | 0.834569 | 0.895615 | 0.576171 | 0.654832 | 0.463726 | 0.467382 |
| BZ2 | 0.662966 | 0.497389 | 0.59459 | 0.870364 | 0.559904 | 0.679875 | 0.396984 | 0.521292 |
| ERROR | 0.801987 | 0.696275 | 0.897926 | 0.896308 | 0.598028 | 0.682492 | 0.527293 | 0.608899 |
| BLOCKED | 0.687398 | 0.696601 | 0.772956 | 0.850647 | 0.596924 | 0.49704 | 0.520056 | 0.600729 |
| CONTINUITY | 0.849582 | 0.791413 | 0.88805 | 0.944203 | 0.678348 | 0.802402 | 0.59703 | 0.71316 |
| POINT | 0.855345 | 0.780546 | 0.889369 | 0.93668 | 0.670617 | 0.769469 | 0.569567 | 0.698489 |

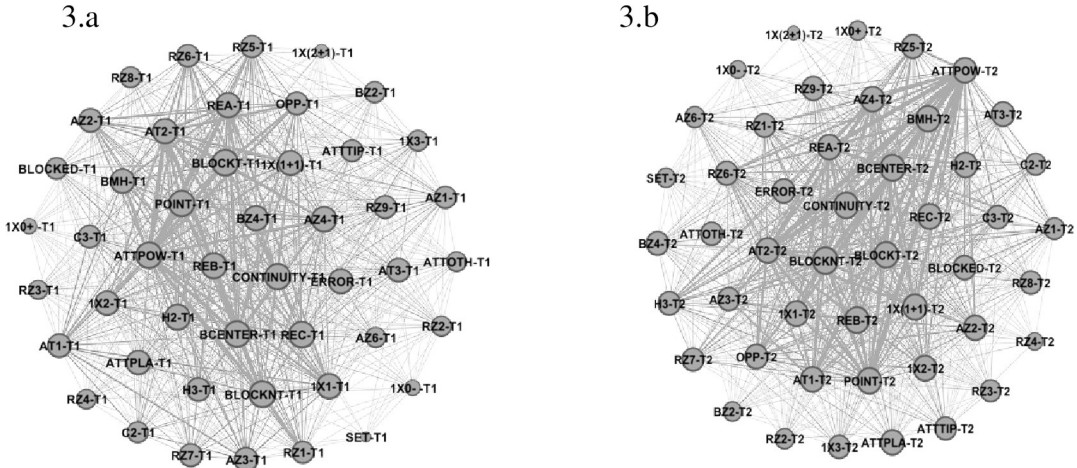

**Fig 3.** Sub-network with intra-team variability of Team 1 (3.a) and 2 (3.b).

and Point (0.85). Block Structure Centralized (0.87) had the highest centrality values, while 1x1 and 1x2 the same value (0.82 for both) and Block not Touch (0.91). Team 4 (Serbia) revealed higher values (See Table 1 and Fig 4B) in the following variables: Reception Zone 8 and 9 (0.83 for both) Reception Effect A (0.97) and C (0.96), Attack Zone 4 (0.90) and 2 (0.88), Attack Tempo 3 (0.95) and 2 (0.91), with outside Hitter 2 (0.92) and Outsider Hitter 3 (0.89), Powerful Attack (0.94) and Attack Effect Continuity (0.94) and Point (0.93). The variables related to blocking presented the following centrality values: Block Structure Centralized (0.95), while 1x(1+1) and 1x2 with the same value (0.89 for both) and Block Not Touch (1.0).

The fifth place in the competition (Team 5—Italy) presented higher eigenvector values (See Table 1 and Fig 5A) for Reception Zone 6 (0.53) and 5 (0.52), Reception Effect A (0.59) and C (0.57), Attack Zone 4 (0.58) and Attack Tempo 2 (0.56) and 1 (0.53), Opposite (0.55), Powerful Attack (0.59) and Attack Effect Continuity (0.58) and Point (0.57). Block Structure Centralized (0.58), with 1x1 (0.56), Block Not Touch (0.60) had higher values in the categories in KI. Team 6 (Russia) revealed higher centrality values (See Fig 5B) in the following variables: Reception Zone 6 (0.64), Reception Effect C (0.73), Attack Zone 4 (0.71) and 2 (0.64), Attack Tempo 2

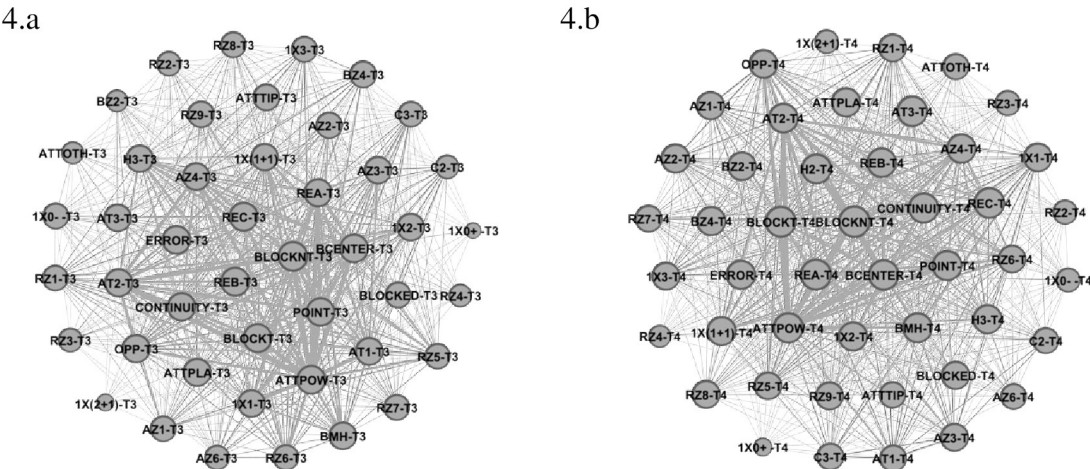

**Fig 4.** Sub-network with intra-team variability of Team 3 (4.a) and 4 (4.b).

5.a

5.b

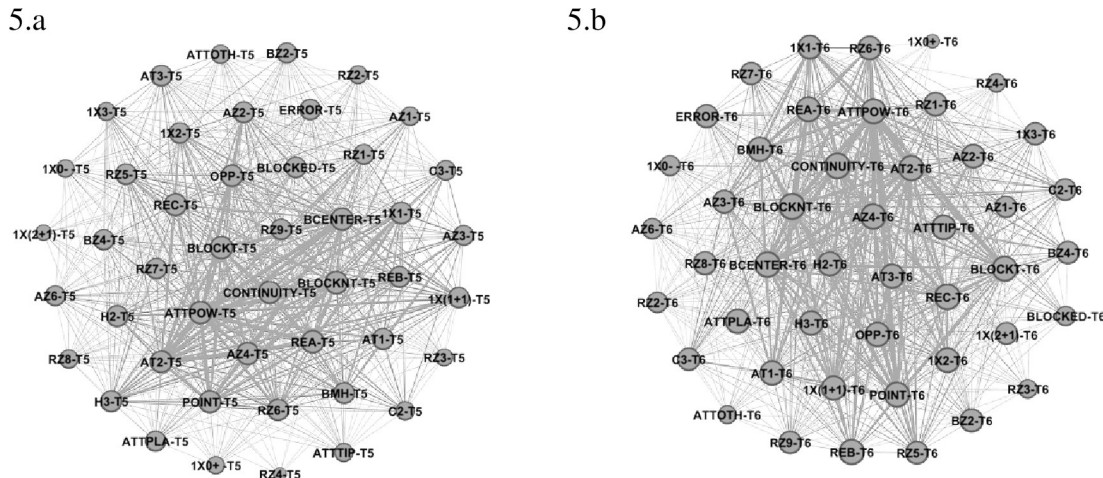

**Fig 5.** Sub-network with intra-team variability of Team 5 (5.a) and 6 (5.b).

(0.70) and 1 (0.67), with Opposite (0.67), Powerful Attack (0.70) and Attack Effect Continuity (0.73) and Point (0.70). The variables related to blocking presented higher values to Block Structure Centralized (0.72), Composition 1x2 (0.67) and 1x(1+1) (0.66) and Block Not Touch (0.74).

Team 7 (France) showed higher centrality values (See Table 1 and Fig 6A) for Reception Zone 6 and 1 (0.43 for both), with Reception Effect B (0.48) and C (0.46), Attack Zone 4 (0.45) and Attack Tempo 3 (0.45) and 2 (0.45), Outside Hitter 2 (0.45), Placed Attack (0.48), Attack Effect Continuity (0.48) and Point (0.45). Block Structure Centralized (0.48) had the highest centrality values, while Block Composition 1x2 (0.45) and 1x1 (0.44) and Block not Touch (0.49) had the highest Eigenvector values. Team 8 (Holland) revealed higher values (See Table 1 and Fig 6B) in the following variables: Reception Zone 5 (0.55), Reception Effect C (0.57) and B (0.58), Attack Zone 4 (0.58) and 3 (0.56), Attack Tempo 2 (0.58) and 1 (0.54), with Opposite (0.55), Powerful Attack (0.60) and Attack Effect Continuity (0.60) and Point (0.59). The variables related to blocking presented the following centrality values: Block Structure Centralized (0.62), while 1x1 (0.57) and 1x2 (0.54) and Block not Touch (0.63).

6.a

6.b

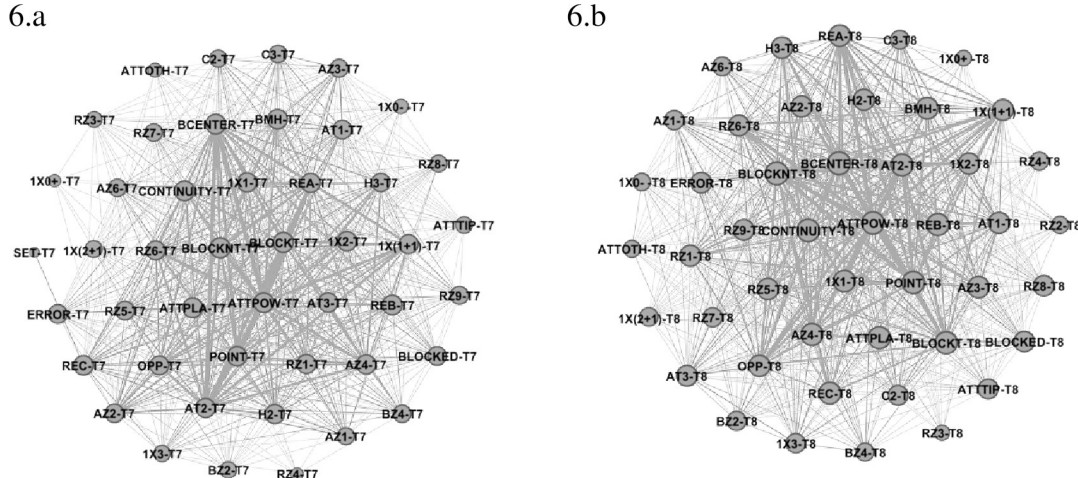

**Fig 6.** Sub-network with intra-team variability of Team 7 (6.a) and 8 (6.b).

## Discussion

Social Network Analysis was used to analyze the connection and specificity [13] in Complex I, considering the direct and indirect connections between nodes through the Eigenvector Centrality [10, 53]. The present study purposes were: (a) to analyze at a macro-level the important variables for the construction of the KI of the eight best ranked teams that participated in the 2018 Volleyball World Championship through a single and global competition network, and (b) perform micro-analysis considering the peculiarities (differences and similarities) in the offensive construction in KI that distinguish the multiple performance models of the teams participating in the competition.

The first hypothesis that: in macro-analysis view the teams will present higher centrality values to excellent setting conditions (reception effect A), attack tempo 1, attack from zone 3, against simple block (1x1), with attack does not touch block and attack effect point was partially confirmed. The results showed that the highest eigenvector values were found for reception zone 9, reception effects A and C, attack tempo 3 and 2, powerful attack, outside-hitter H2, attack through zone 4 and 2, blocks 1x1 and 1x2, attacks that did not touch the block, middle-blocker centralized to react to the setting, attack effect point and continuity. Research in volleyball match analysis indicates that, when considering complex 1, teams play "in the system", that is, with reception that allow an organized attack, faster attack tempo, attacks from net extremities (zone 4 and 2), powerful attacks performed against less defensive structuring and attack effect point [24, 27, 45].

However, the Complex I has greater velocity in offensive structure than Complex II [24, 53], requiring teams to take more risks in serve with more power, in order to limit this offensive construction, mainly to difficult the increase the attack tempo 1 [18, 54]. In this context, high-level teams are likely to be able to serve with power and in vulnerable locations, a fact that provided high eigenvector values for the reception effect C, requesting the play off-system. In addition, it is noted that the settings at the net extremities are requested, suggesting that teams have a traditional game pattern, making it necessary to vary the setting, mainly with settings for the central court areas (zone 3 and zone 8), increasing unpredictability and difficulting the structuring the defensive system. In this context, the macro analysis indicated that, although the teams show serve strategies that aim to difficult the offensive structuring, the game type is similar to research carried out in recent decades [45, 54, 55].

The second hypothesis was the micro-analysis the top 2 ranked teams will present higher eigenvector values in the reception effect, attack and blocking effectivity was partially confirmed. The reception analysis showed that the 1st to 4th placed teams had higher eigenvector values for receptions from zone 9, the 5th to 7th placed teams to zone 6 and the 8th placed team to zone 5. In addition, the 4th and 5th places teams presented higher eigenvector values for REA in relation to the other reception effects, while the other teams presented higher values for REB and REC. The reception micro-analysis confirms the game trend existing in high-level volleyball, in which it is necessary that the serve restricts the offensive construction [18, 46]. Therefore, directing the serve in more distant zones of the net and in the net outside hitter showed a decrease in the offensive structuring, as well as reducing the reception effect [22, 43].

In addition, directing the serve in the zones closest to zones 9 and 1, since the setter performs the distribution from balls received in the opposite direction to the displacement performed, that is, settings that come from the "setter back", makes it difficult the perception of the blockers disposition, especially in critical game scenarios [31, 33, 38]. The attack tempo analysis showed that the teams presented higher eigenvector values for attack tempo 2, except for the 4th and 7th placed teams that presented higher eigenvector values for attack tempo 3.

When considering the variables related to the attack, it can be seen that the attack type, for the most part, presented higher centrality values for: the powerful attack with the exception of

the 7th placed team with the placed attack; play position outsider hitter—H2 for the 1st, 2nd, 4th and 7th teams, while the others for the OPP; and attacks performed by zone 4 for all teams. Although the literature arguments that attack tempo 1 and 2, attacks by middle blockers and powerful attacks are better for offensive structuring [25, 56, 57], there was a lower tendency of the faster attacks and the game played by the net zone extremities. This fact suggests that teams are dependent on security attackers, that is, those with power in non-ideal attack situations, as is the case of the opposite player [19, 58]. In addition, there is the possibility that teams prefer, in non-ideal attack conditions, to play with the attack coverage system, allowing the recovery of the attacked balls and the offensive reconstruction (For more information, read: Laporta, Nikolaidis [12], Laporta, Afonso [14], Laporta, Afonso [24], Hurst, Loureiro [59]). In this context, teams use the attack coverage system to achieve better conditions for the offensive construction, reducing the risk of error that emanates from the powerful attack in restricted conditions of action possibilities [12, 60].

Eigenvector values for the variables associated to the blocking, it was observed that: the 1st, 3rd, 5th and 8th placed teams presented higher eigenvector values for the 1x1 blocking, and the 2nd for the 1x(1+1) situation and the other teams for the 1x2 situation; all teams presented higher eigenvector values for balls that did not touch the block and for the Middle-blocker disposition waiting to react to the setting location. These results corroborate the literature, since the blocks faced are mostly single or double [61, 62]. On the other hand, the need to avoid the block touching in attacked ball is notorious, since this touch reduces the attack speed, allowing more time for the defense reaction and subsequent counterattack construction [22, 43]. In addition, the blocking strategy of reacting to the ball prevents the attack from overlapping, since, in conditions of organized attack, it is possible to use 4 attackers in the current offensive structure of high-level teams [55].

Related to the attack effect, it was observed that the 1st placed team had higher eigenvector values for the attack point, the 3rd placed team for the attack error and the other teams showed higher eigenvector values for the attack continuity. The results corroborate the current literature, since the attack points distinguish the chances of winning the set and the game [21, 63]. The volleyball game claims the attack with power in vulnerable areas of the opposing team, limiting the possibilities of defensive organization [19, 25]. Therefore, teams must be able to adapt the offensive game type to the restrictions imposed by the opposing defense, with unpredictability in offensive actions being crucial, limiting the time for defensive system adaptations [57, 64]. Thus, the champion team of the present study managed to overcome the opposing system, indicating that it did not allow the adaptation of the opposing defensive system to the possibilities of offensive action.

## Conclusion

A global network showing intra- and inter-team variability was established through interactions between the eight teams, however, we took an approach that respected the idiosyncrasies of each competing team, thus creating a model that considers each team rather than an aggregate model using the SNA. We are convinced that exposing how different high-level teams can be effective using different approaches to the game will provide coaches with a broader understanding of achieving success possibilities and will invite coaches to explore their own team skills rather than copying a standardized model, also because the individual players characteristics can vary dramatically at elite levels [65]. In this way, we consider the constant and dynamic game changes [10] within its ecological context [35]. Evidencing multiple paths of analysis is important because, as seen in our results, network analysis demonstrate differences in play patterns, with different distributions of their centrality values. This should be

highlighted because volleyball is a sport in which game sequences do not offer as much variability as other team sports, because of rules that limit the number of contacts per ball possession.

Although the results are applicable and represent the game type played by the world best teams, the present research has limitations, as the fact that we do not analyze the inter- and intra-team variability according to the confrontation carried out, restricts the extrapolation power of the results, since different strategies can be used game by game, as well as the team situation in the competition can interfere in the adopted game pattern. Therefore, we suggest that future studies analyze the tactical behavior of teams in each game, seeking to find patterns that specify team performance, since the elite men's teams have a tendency to play in-system in KI situations [14, 15, 24], it is necessary for coaches carry out training in- and off-system [22], being important to prepare the teams for the competition and develop the ability to respond to the constraints imposed by the game's ecology, considering the critical game scenarios [38, 66].

## Supporting information

**S1 Data.**
(XLSX)

## Author Contributions

**Conceptualization:** Thiago José Leonardi, Henrique de Oliveira Castro, Gustavo De Conti Teixeira Costa.

**Data curation:** Geovana Pires Rodrigues.

**Formal analysis:** Augusto Cezar Rodrigues Rocha, Geovana Pires Rodrigues, Marcelo Couto Jorge Rodrigues, Claudio Andre Barbosa de Lira.

**Investigation:** Augusto Cezar Rodrigues Rocha, Lorenzo Laporta, Juracy da Silva Guimarães, Marcos Henrique do Nascimento, Marcelo Couto Jorge Rodrigues, Gustavo De Conti Teixeira Costa.

**Methodology:** Augusto Cezar Rodrigues Rocha, Lorenzo Laporta, Geovana Pires Rodrigues, Marcelo Couto Jorge Rodrigues, Thiago José Leonardi, Henrique de Oliveira Castro.

**Project administration:** Marcos Henrique do Nascimento, Thiago José Leonardi, Henrique de Oliveira Castro.

**Resources:** Juracy da Silva Guimarães, Henrique de Oliveira Castro.

**Software:** Marcelo Couto Jorge Rodrigues, Claudio Andre Barbosa de Lira.

**Supervision:** Marcos Henrique do Nascimento, Thiago José Leonardi, Claudio Andre Barbosa de Lira, Gustavo De Conti Teixeira Costa.

**Validation:** Marcelo Couto Jorge Rodrigues, Claudio Andre Barbosa de Lira.

**Visualization:** Juracy da Silva Guimarães, Henrique de Oliveira Castro.

**Writing – original draft:** Lorenzo Laporta, Juracy da Silva Guimarães, Marcos Henrique do Nascimento, Thiago José Leonardi, Gustavo De Conti Teixeira Costa.

**Writing – review & editing:** Lorenzo Laporta, Claudio Andre Barbosa de Lira, Henrique de Oliveira Castro, Gustavo De Conti Teixeira Costa.

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
