## [Decision Letter · Decision Letter 0]

12 Oct 2022

PONE-D-22-23691Inter and intra-variability of the best ranked teams: a network analysis in male high-level VolleyballPLOS ONE

Dear Dr. Costa,

Thank you for submitting your manuscript to PLOS ONE. After careful consideration, we feel that it has merit but does not fully meet PLOS ONE’s publication criteria as it currently stands. Therefore, we invite you to submit a revised version of the manuscript that addresses the points raised during the review process.

 Despite the exciting research topic, the three reviewers highlighted that major modifications should come in methods and discussion. It is strongly recommended to strictly follow the reviewers' recommendations for improving the replicability and overall quality of the article.

We look forward to receiving your revised manuscript.

Kind regards,

Filipe Manuel Clemente, PhD

Academic Editor

PLOS ONE

Journal Requirements:

3. Please ensure that you have specified (1) whether consent was informed and (2) what type you obtained (for instance, written or verbal, and if verbal, how it was documented and witnessed). If your study included minors, state whether you obtained consent from parents or guardians. If the need for consent was waived by the ethics committee, please include this information.

"NO authors have competing interests"

5. Please upload a copy of Figure S1-S5, to which you refer in your text on page xx. If the figure is no longer to be included as part of the submission please remove all reference to it within the text.

Reviewers' comments:

Reviewer's Responses to Questions

**Comments to the Author**

1. Is the manuscript technically sound, and do the data support the conclusions?

Reviewer #1: Yes

Reviewer #2: Partly

Reviewer #3: Yes

2. Has the statistical analysis been performed appropriately and rigorously? 

Reviewer #1: Yes

Reviewer #2: No

Reviewer #3: Yes

3. Have the authors made all data underlying the findings in their manuscript fully available?

Reviewer #1: Yes

Reviewer #2: No

Reviewer #3: No

4. Is the manuscript presented in an intelligible fashion and written in standard English?

Reviewer #1: Yes

Reviewer #2: No

Reviewer #3: No

5. Review Comments to the Author

Reviewer #1: First of all, I would like to congratulate the authors for their work. I send you some improvement suggestions.

- In the introduction the problem must be presented clearly. Despite being referred to in a general way, I suggest that the problem is presented specifically.

- Does it matter in the sample whether the observed matches correspond to the same stages of the competition? If not, I think it is important to clarify this situation and consider presenting results according to this contextual variable. The observed actions took into account contextual variables (eg opponent level), a fact that can lead to different presentations from those recorded.

- The variables presented are part of an observational instrument. Did you go through any validation process?

- It should also be mentioned how the training of the observers was carried out.

- What are the procedures for testing inter- and intra-observer reliability?

- Reliability values per variable must be presented.

- I suggest the presentation of practical implications and the main conclusions of the study

Reviewer #2: I would like to thank the editors for giving me the opportunity to read this interesting contribution.

This contribution has the objective to analyze the offensive actions from the Complex I phase in volleyball by studying the network structure concerning inter- and intra-team relationships of the 2018 men's volleyball world championship data.

This manuscript puts forward an interesting idea, but it lacks a proper methodological formalization.

It is not clear how the networks have been constructed and the methodological part regarding the analysis needs to be addressed properly. In particular, the authors need to specify what kind of ties they are considering when constructing the network, highlighting what are the advantages of choosing the eigenvector centrality compared of other centrality indices. In addition, although the overall English level is acceptable, a thorough spell check is necessary.

I suggest the authors to resubmit the manuscript after adding a more consistent methodological part.

Reviewer #3: Despite the work carried out, the writing in English is very weak. Consequently, reading becomes difficult to understand. The Introduction does not have a logical sequence of thoughts and the Discussion can be improved according to the studies already carried out. Only with a major review of these two chapters, one could consider accepting the paper.

6. PLOS authors have the option to publish the peer review history of their article (what does this mean?). If published, this will include your full peer review and any attached files.

Reviewer #1: **Yes: **Fernando Santos

Reviewer #2: No

Reviewer #3: No

---

## [Author Response · Author response to Decision Letter 0]

15 Nov 2022

Response to reviewers

General Comments

1. Please ensure that your manuscript meets PLOS ONE's style requirements, including those for file naming 

We modified all the text as PLOS ONE’s Style.

2. We suggest you thoroughly copyedit your manuscript for language usage, spelling, and grammar. 

We have proofread the entire text.

3. Please ensure that you have specified (1) whether consent was informed and (2) what type you obtained (for instance, written or verbal, and if verbal, how it was documented and witnessed). If your study included minors, state whether you obtained consent from parents or guardians. If the need for consent was waived by the ethics committee, please include this information.

Information on ethics committee approval is available in the methods section.

"NO authors have competing interests"

No authors have competing interests

5. Please upload a copy of Figure S1-S5, to which you refer in your text on page xx. If the figure is no longer to be included as part of the submission please remove all reference to it within the text.

 Done

Reviewers' comments:

3. Have the authors made all data underlying the findings in their manuscript fully available?

We have added all data in the submission.

- In the introduction the problem must be presented clearly. Despite being referred to in a general way, I suggest that the problem is presented specifically.

We put the problem more clearly.

- Does it matter in the sample whether the observed matches correspond to the same stages of the competition? If not, I think it is important to clarify this situation and consider presenting results according to this contextual variable. The observed actions took into account contextual variables (eg opponent level), a fact that can lead to different presentations from those recorded.

The matches observed were of the same opponent level, this information was added directly to the text:

“A total of 2,743 offensive actions in Complex I were analyzed, from 22 matches of 2018 Men's Volleyball World Championship between the eight best ranked teams in this edition (Poland, Brazil, United States, Serbia, Italy, Russia, France and Netherlands respectively).”

- The variables presented are part of an observational instrument. Did you go through any validation process? We utilized variables already validated and used in other studies (as referenced in each description).

- It should also be mentioned how the training of the observers was carried out. 

- What are the procedures for testing inter- and intra-observer reliability? 

We followed the reviewer's recommendations directly in the text:

“The analysis was performed by three volleyball coaches (more than five years of experience as performance analysts), trained by a high-level coach with more than 10 years of practice and with experience with national and regional teams. For this purpose, 5 games were analyzed together from a competition different from the sample (Final phase of the Men's Brazilian Superliga 20/21), and the final game (composed of 5 sets) was reanalyzed after 1 month to verify the intra and inter-observer reliability, thus resolving any doubts. For the final reliability test, 20% of the actions were reanalyzed, which is above the reference value of 10% [47]. Cohen's Kappa values were between 0.89 and 0.98 with the respective standard errors of 0.07 and 0.01 for the intraobserver analysis, and 1 with the standard error equal to 0 for the interobserver analysis. Such values are above the value recommended by the literature, which is 0.75 [48].”

- Reliability values per variable must be presented. 

We added directly to the text.

Reliability values for each variable were: Reception Zone (0,95), Reception Effect (0,89); Attack Tempo (0,95), Attack Type (0,98), Attack Zone (1,00), Play Position (1,00), Block Composition (0,98), Block Touch (0,98), and Attack Effect (0,98)

- I suggest the presentation of practical implications and the main conclusions of the study Done

It is not clear how the networks have been constructed and the methodological part regarding the analysis needs to be addressed properly. In particular, the authors need to specify what kind of ties they are considering when constructing the network, highlighting what are the advantages of choosing the eigenvector centrality compared of other centrality indices. 

Methodological process was better explained directly in the text.

“Thus, the eigenvector centrality provides the relevant information about which nodes, or here as game variables used, are more influential in the network, taking into account the connectivity from other nodes that are also more central [49, 50], so, the Eigenvector Centrality depends not only on the number of its adjacent nodes, but also their interaction characteristics [40]. Node size were manipulated to highlight the magnitude of the eigenvector measure using the intrinsic units provided by Gephi Software (Between 300 to 1,500 arbitrary units). Edge corresponds the direct relationship between two nodes defined by number of connections, therefore, thicker edges correspond to a greater number of connections between two nodes [51]. Thus, the node size determines the visual variables contrast according to the eigenvector centrality and the edges thickness, in turn, reveals the weight (given by the number) of the connection directly and indirectly between the nodes [10, 24]. In addition, the Modularity Algorithm was used to detect community structure (8 communities with a cohesion value of 0.871) and the "Fruchterman Reigold" distribution (area 100000) was used organizing the nodes with the highest eigenvector centrality in the center of each subnet [52].”

I suggest the authors to resubmit the manuscript after adding a more consistent methodological part. 

Following the recommendations of Reviewer #1, we changed and improved the methodological process of every session directly in the text

---

## [Decision Letter · Decision Letter 1]

6 Dec 2022

PONE-D-22-23691R1Inter and intra-variability of the best ranked teams: a network analysis in male high-level VolleyballPLOS ONE

Dear Dr. Costa, Thank you for submitting your manuscript to PLOS ONE. After careful consideration, we feel that it has merit but does not fully meet PLOS ONE’s publication criteria as it currently stands. Therefore, we invite you to submit a revised version of the manuscript that addresses the points raised during the review process.

Please take a look and correct the references (e.g. 21, 23, and the underlining DOI of the references).

Additionally, you should add figure in the reception zone variable (RZ).

We look forward to receiving your revised manuscript.

Kind regards,

Filipe Manuel Clemente, PhD

Academic Editor

PLOS ONE

Journal Requirements:

Reviewers' comments:

Reviewer's Responses to Questions

**Comments to the Author**

1. If the authors have adequately addressed your comments raised in a previous round of review and you feel that this manuscript is now acceptable for publication, you may indicate that here to bypass the “Comments to the Author” section, enter your conflict of interest statement in the “Confidential to Editor” section, and submit your "Accept" recommendation.

Reviewer #1: All comments have been addressed

Reviewer #3: All comments have been addressed

2. Is the manuscript technically sound, and do the data support the conclusions?

Reviewer #1: Yes

Reviewer #3: Yes

3. Has the statistical analysis been performed appropriately and rigorously? 

Reviewer #1: Yes

Reviewer #3: Yes

4. Have the authors made all data underlying the findings in their manuscript fully available?

Reviewer #1: Yes

Reviewer #3: Yes

5. Is the manuscript presented in an intelligible fashion and written in standard English?

Reviewer #1: Yes

Reviewer #3: Yes

6. Review Comments to the Author

Reviewer #1: The authors performed a review of the article as recommended by the reviewer. In my opinion the article is ready to be published.

Reviewer #3: Dear Authors! First of all, congratulations for your effort regarding this second submission. In my opinion, the purpose of the study is adequately to the results and the methodology is now clear, according to previous sugestions. Please take a look and correct the references (e.g. 21, 23, and the underlining DOI of the references).

Additionally, you should add figure in the reception zone variable (RZ).

7. PLOS authors have the option to publish the peer review history of their article (what does this mean?). If published, this will include your full peer review and any attached files.

Reviewer #1: **Yes: **Fernando Santos

Reviewer #3: No

---

## [Author Response · Author response to Decision Letter 1]

12 Dec 2022

Dear Editor in Chief, 

We submit an original research article entitled “Inter e intra variability of the best ranked teams: an network analysis in high-level Volleyball” for consideration of the PLOS ONE. The reviewer 3 requested corrections in the references, as well as the insertion of a figure with the reception zones. We complied with the reviewer's request. We emphasize that the DOI was in blue, as it is an access link. In addition, we entered the DOI of all manuscripts that contained this identification.

Thank you for your consideration of this manuscript.

---

## [Decision Letter · Decision Letter 2]

21 Dec 2022

PONE-D-22-23691R2Inter e intra variability of the best ranked teams: a network analysis in high-level VolleyballPLOS ONE

Dear Dr. Costa,

Thank you for submitting your manuscript to PLOS ONE. After careful consideration, we feel that it has merit but does not fully meet PLOS ONE’s publication criteria as it currently stands. Therefore, we invite you to submit a revised version of the manuscript that addresses the points raised during the review process.

Please consider the latest comments from reviewer 1

We look forward to receiving your revised manuscript.

Kind regards,

Filipe Manuel Clemente, PhD

Academic Editor

PLOS ONE

Journal Requirements:

Reviewers' comments:

Reviewer's Responses to Questions

**Comments to the Author**

1. If the authors have adequately addressed your comments raised in a previous round of review and you feel that this manuscript is now acceptable for publication, you may indicate that here to bypass the “Comments to the Author” section, enter your conflict of interest statement in the “Confidential to Editor” section, and submit your "Accept" recommendation.

Reviewer #1: (No Response)

Reviewer #3: All comments have been addressed

2. Is the manuscript technically sound, and do the data support the conclusions?

Reviewer #1: Yes

Reviewer #3: Yes

3. Has the statistical analysis been performed appropriately and rigorously? 

Reviewer #1: Yes

Reviewer #3: Yes

4. Have the authors made all data underlying the findings in their manuscript fully available?

Reviewer #1: Yes

Reviewer #3: Yes

5. Is the manuscript presented in an intelligible fashion and written in standard English?

Reviewer #1: Yes

Reviewer #3: Yes

6. Review Comments to the Author

Reviewer #1: The authors answered the vast majority of the questions raised, however I leave two notes:

- There is still spacing in the references after the comma.

- With regard to the observational instrument, it is not clear whether or not there was a validation process. It is noticed that the variables were listed according to the literature review, but validation by experts is not evident, or another type of instrument validation was performed. I suggest you see the following bibliographical reference:

Pulido, J.J., Sánchez-Oliva, D., Silva, M.N., Palmeira, A.L., & García-Calvo, T. (2019). Development and preliminary validation of the Coach interpersonal Style Observational System. International Journal of Sports Science & Coaching, 14(4), 471-479

Reviewer #3: Dear Authors,

Thank you for your effort. All recommendations have been made according the suggestions. In my opinion, the paper could be accepted in this final version.

7. PLOS authors have the option to publish the peer review history of their article (what does this mean?). If published, this will include your full peer review and any attached files.

Reviewer #1: **Yes: **Fernando Jorge Santos

Reviewer #3: No

---

## [Author Response · Author response to Decision Letter 2]

27 Dec 2022

Dear Reviewer 1, 

We have submitted an original research article entitled “Inter e intra variability of the best ranked teams: a network analysis in high-level Volleyball” for consideration of the PLOS ONE. The reviewer 1 requested corrections in the references, as well as more information on the analysis of coaches. The extra spaces in the references section has been removed. In the methods section, we added the following information: The coaches used the Data Volley software to control and analyze the scenes. Each analyzed dimension was discussed by the coaches, ensuring that the categories were exclusive and exhaustive to represent the different game scenarios, as well as that there was a unanimous consensus among all evaluators as suggested by Pulido et al. [47]. We emphasize that we inserted the reference suggested by the reviewer. We complied with the reviewer's request. 

Thank you for your consideration of this manuscript. 

Best Regards.

---

## [Editor Report · Decision Letter 3]

28 Dec 2022

Inter e intra variability of the best ranked teams: a network analysis in high-level Volleyball

PONE-D-22-23691R3

Dear Dr. Costa,

We’re pleased to inform you that your manuscript has been judged scientifically suitable for publication and will be formally accepted for publication once it meets all outstanding technical requirements.

Kind regards,

Filipe Manuel Clemente, PhD

Academic Editor

PLOS ONE
---

## [Editor Report · Acceptance letter]

12 Jan 2023

PONE-D-22-23691R3 

Inter e intra-variability of the best ranked teams: a network analysis in male high-level Volleyball. 

Dear Dr. Costa:

I'm pleased to inform you that your manuscript has been deemed suitable for publication in PLOS ONE. Congratulations! Your manuscript is now with our production department. 

Kind regards, 

on behalf of

Dr. Filipe Manuel Clemente 

Academic Editor

PLOS ONE